# Effects of Graphene Derivatives and Near-Infrared Laser Irradiation on *E. coli* Biofilms and Stress Response Gene Expression

**DOI:** 10.3390/ijms26104728

**Published:** 2025-05-15

**Authors:** Yuliya Maksimova, Ekaterina Pyankova, Larisa Nesterova, Aleksandr Maksimov

**Affiliations:** 1Laboratory of Molecular Biotechnology, Institute of Ecology and Genetics of Microorganisms UB RAS, 614081 Perm, Russia; 19katya991@rambler.ru (E.P.); almaks1@mail.ru (A.M.); 2Department of Microbiology and Immunology, Perm State University, 614990 Perm, Russia; 3Laboratory of Microbial Adaptation, Institute of Ecology and Genetics of Microorganisms UB RAS, 614081 Perm, Russia; larisa.nesterova@bk.ru; 4Department of Plant Physiology and Soil Ecology, Perm State University, 614990 Perm, Russia

**Keywords:** biofilms, carbon nanomaterials, graphene oxide, reduced graphene oxide, oxidative stress, near-infrared laser, photothermal therapy, stress response gene expression, extracellular polymeric substances

## Abstract

Photothermal therapy combines the effects of near-infrared laser (NIR laser) and strong light-absorbing materials to combat pathogens and unwanted biofilms. Graphene derivatives have a negative effect on microorganisms, and the combination of NIR laser irradiation and carbon nanomaterials (CNMs) can enhance their antibacterial effect. This investigation is devoted to the determination of the expression level of bacterial stress response genes (*soxS* and *rpoS*) under graphene oxide (GO), reduced graphene oxide (rGO), and NIR laser irradiation (1270 nm). GO, rGO and NIR laser irradiation separately and irradiation in the presence of graphene derivatives cause an increase in the expression level of *rpoS* associated with the general stress response of bacteria. GO and rGO do not change the expression level of *soxS* associated with the cell response to oxidative stress, and decrease it in the presence of a strong oxidizing agent paraquat (PQ). The expression of *soxS* increases under laser irradiation, but decreases under NIR laser irradiation in combination with graphene derivatives. The effect of GO, rGO, and NIR laser irradiation on the formation and eradication of *E. coli* biofilms was studied. NIR laser with GO and rGO suppresses the metabolic rate and decreases the intracellular ATP content by 94 and 99.6%, respectively. CNMs are shown to reduce biofilm biomass and the content of extracellular polymeric substances (EPSs), both exopolysaccharides and protein in the biofilm matrix. Graphene derivatives in combination with NIR laser irradiation may be an effective means of combating emerging and mature biofilms of Gram-negative bacteria.

## 1. Introduction

Graphene is an allotropic modification of carbon, consisting of a one-atom-thick layer of carbon. The carbon atoms in graphene are in sp^2^-hybridization. Graphene oxide (GO) is obtained via the oxidation of graphite flakes. GO contains oxygen functional groups, namely hydroxyl, carboxyl, carbonyl and epoxy groups. Reduced graphene oxide (rGO) is obtained via the chemical or thermal reduction of oxygen in GO [1,2]. Graphene has unique chemical and physical (optical, electrical, mechanical) properties and a large surface area. It is widely used in electronics, instrumentation, space technology and, in addition, has potential applications in biomedicine, such as in biosensors, cancer therapy, targeted drug therapy and antimicrobial treatment [3]. The antimicrobial and antifouling properties of graphene nanomaterials have been widely reported [4]. Graphene nanomaterials are used in water purification [5], medicine [6], biosensors [7], and anti-biofilm materials [8]. The antibacterial activity of graphene, its derivatives, and graphene-based materials is reported to be due to direct physical and chemical interactions between graphene and microbial cells. Graphene causes the degradation of cellular components, mainly proteins, nucleic acids, and lipids, and accumulates in membrane proteoglycans, causing membrane damage. Graphene penetrates the protein–protein interface of the dimeric protein and destabilizes protein–protein interactions by disrupting hydrophobic bonds, leading to the decomposition of the protein complex. In addition, graphene nanomaterials interrupt the replication phase by interacting with hydrogen groups of bacterial RNA/DNA [9].

The formation of reactive oxygen species (ROS) and the development of oxidative stress are considered one of the main reasons for the antibacterial properties of graphene [10]. ROS are formed in the environment or in the cell under the influence of various chemical or physical stress factors. The amount of ROS is maintained at a low level under normal conditions due to the action of antioxidant systems. The *oxyR* and *soxRS* regulons protect cells from hydrogen peroxide and superoxide radicals, respectively [11]. In the SoxRS-regulating system, SoxR operates as a sensor and SoxS as a regulator, controlling the transcription of about 100 genes encoding proteins that protect cells from the damaging effects of ROS [12]. In addition to oxidative stress protection systems, one of the main regulators of the stress response in bacterial cells is RpoS, the sigma subunit of RNA polymerase, which is induced by various stress factors. More than 500 genes (about 10% of the genome) are controlled by this alternative sigma subunit, and their products represent the metabolic arsenal of the cell in non-optimal conditions [13]. A high level of *soxS* and *rpoS* expression in the cell indicates the development of oxidative stress and the general stress response, respectively.

GO, due to the large number of oxygen-containing functional groups (-COOH, -OH, and others) on its surface, can promote the formation of ROS, which increases its antibacterial properties. At the same time, a higher level of oxidative stress is usually found in cells exposed to GO compared to rGO, also due to the high density of GO crystal lattice defects and its higher dispersibility [14].

Although many researchers believe that graphene nanomaterials have antibacterial properties, the data on the antimicrobial and antibiofilm properties of these materials are quite contradictory. Akhavan et al. reported greater resistance to GO and rGO in Gram-negative *E. coli* compared to *Staphylococcus aureus*, as well as the higher toxicity of rGO to bacterial cells [15]. The reduction of GO has been shown to make this material more effective against bacteria than its unreduced counterpart. Due to its greater hydrophobicity, it interacts with the lipid layer of the membrane and causes its destruction [16]. GO significantly enhanced the growth, formation and development of *E. coli* and *S. aureus* biofilms even at a concentration of 500 mg/L, while rGO (≥50 mg/L) showed a strong inhibitory effect on planktonic cells and biofilms, and the inhibitory effect of rGO (50 and 100 mg/L) was decreased by 24 h (mature biofilm phase) and was eliminated after 48 h [17]. In contrast, Liu et al. showed that GO dispersion had higher antibacterial activity than rGO. ROS, including superoxide anion, were not formed, but graphene nanoparticles could oxidize glutathione, which serves as a mediator of the redox state in bacteria [18]. The action of graphene begins with direct contact with the cell, which causes membrane stress and subsequent oxidation independent of the superoxide anion. The concentration of oxygen in graphene derivatives modifies its antimicrobial effect. Saeed et al. reported the antimicrobial activity of GO against bacterial cells. The viability of *S. aureus* was reduced by 90% upon exposure to 200 μg/mL GO. GO also demonstrated antibiofilm activity. Upon exposure to 100 μg/mL GO, the biofilm biomass was reduced by 30–70% [19]. However, low doses of GO, which do not have a bacteriostatic effect, can promote bacterial proliferation and lead to an increase in bacterial populations in the aquatic environment [20]. At low temperatures, GO enhances community metabolism by stimulating the consumption of organic carbon and regulating the expression of genes encoding enzymes of the Krebs and Entner–Doudoroff cycles, which, as a consequence, may stimulate the synthesis and secretion of extracellular polymeric substances (EPSs) [21].

To enhance the effect of graphene nanomaterials on unwanted microorganisms and their biofilms, these carbon nanoparticles in various forms (graphene quantum dots, GO, etc.) can be combined with phototherapy [22,23]. Phototherapy includes photodynamic and photothermal therapy and is used in cancer therapy and antibacterial therapy. ROS are formed in the process of photodynamic therapy when under the effect of a certain wavelength of light, while photosensitizers transfer the energy of photons to oxygen molecules. Under the mechanism of photothermal therapy, materials are able to strongly absorb NIR light and convert it into heat; this causes hyperthermia in cells, tissues or microbial biofilms. NIR laser irradiation has a targeted effect. Light in the NIR range (700–1400 nm) is able to deeply penetrate tissues without damaging them, increasing the probability of bacteria being destroyed in deep tissues. It also enhances the diffusion of antibacterial agents into biofilms through induced hyperthermia. In addition, the mechanism of exposure is non-invasive and contactless, which reduces the probability of developing bacterial resistance, and also leads to few side effects [24,25]. Photothermal therapy can be used to destroy cancer cells and pathogenic bacterial cells. Due to the development of antibiotic resistance in bacteria, this method is becoming increasingly relevant for combating microorganisms resistant to antibiotics [26,27,28]. Photothermal therapy is an alternative to antibiotic therapy in certain cases and is used, for example, in the treatment of infections associated with implantation [29]. Agents that enhance the antibacterial effect of NIR radiation can be various photosensitizers, such as composites based on Prussian blue and silver nanoparticles [30], pentafluorophenyl bacteriochlorin [31], nanocomposites in the form of conjugates of diethylthiatricarbocyanine iodide with gold, molybdenum sulfide and hyaluronic acid [32], nickel-based complexes [33], titanium oxide [34], graphitic carbon nitride [35], and others. For example, molybdenum-disulfide-functionalized polyethylene glycol nanoparticles have peroxidase activity, generating ^●^OH from hydrogen peroxide, and, together with near-infrared radiation, perform the effective destruction of ampicillin-resistant *E. coli* and endospore-forming *Bacillus subtilis*. Hyperthermia induced by 808 nm irradiation can accelerate the oxidation of intracellular glutathione in the presence of nanoparticles, which negatively affects the protection of cells from oxidative stress [36]. In this case, light energy is effectively converted into thermal energy, and nanoparticles are a conductor of this thermal energy. As a result, local hyperthermia occurs, and this causes the destruction of microorganisms and their biofilm components. CNMs, in particular graphene, are capable of NIR optical absorption and high photothermal conversion, which makes them promising agents for photothermal therapy [22,23,37,38]. In addition to excellent light absorbing characteristics, graphene has the following properties that make it a promising agent for the photothermal therapy of bacterial infections: (1) Graphene can be easily modified chemically; (2) Graphene exhibits biocompatibility; (3) Graphene sheets provide good contact with bacterial cells and biofilms due to their 2D structure and large specific surface area [23].

Despite the active study of these issues, the molecular basis of such an effect is not fully understood. The expression of stress response genes has not been studied either under the influence of graphene nanomaterials or under the combined influence of NIR laser irradiation and graphene derivatives. Data on the effect of GO and rGO on planktonic microbial cells and biofilms are contradictory. In this regard, the aim of this study was to determine the expression level of bacterial cell stress response genes (*soxS,* oxidative stress response, and *rpoS*, general stress response) on genetically modified *E. coli* strains harboring gene fusions under GO, rGO and NIR laser irradiation, as well as to study the combined effect of NIR laser and graphene nanomaterials on the formation and eradication of *E. coli* VKM B-3858D biofilms, metabolic intensity and the permeability of the cytoplasmic membrane.

## 2. Results

### 2.1. Effect of Carbon Nanomaterials and NIR Laser Irradiation on the Expression of the rpoS and soxS Genes

The expression of *rpoS* (the general stress response gene) and *soxS* (the gene included in the oxidative stress response regulon *soxRS*) was studied under the effect of GO, rGO, laser irradiation, combined exposure to NIR laser and graphene derivatives, and heating. It was shown that exposure to GO and rGO in the absence of other factors resulted in an increase in *rpoS* expression, and for GO, it was reliable (Figure 1a). Exposure to NIR laser only and exposure to NIR laser combined with graphene derivatives resulted in an increase in *rpoS* expression (Figure 1b). Heating the cells for 2 min at 45 °C did not result in an increase in *rpoS* expression, whereas its expression significantly increased when cells were heated with graphene derivatives (Figure 1c). The number of CFU/mL did not change significantly during growth in the presence of graphene derivatives, after heating and NIR laser irradiation (Appendix A). The number of CFU/mL decreases slightly only after *E. coli* RO91 is grown for 10 h in the presence of rGO after irradiation with a NIR laser at 5 A for 1 min. But this indicates an even greater increase in *rpoS* expression than reflected in the graph.

The expression of *soxS* did not increase under the effect of GO and rGO (Figure 2a); moreover, GO in the presence of a strong oxidizing agent, PQ significantly reduced the effect of the latter, resulting in a decrease in the level of *soxS* expression (Figure 2b). Exposure to NIR laser irradiation resulted in an increase in the expression of *soxS* compared to non-irradiated cells, but the presence of rGO in the medium significantly reduced the expression of *soxS* compared to that after NIR laser irradiation only (Figure 2c). The number of CFU/mL did not change significantly during growth in the presence of graphene derivatives, PQ and after NIR laser radiation exposure (Appendix A).

### 2.2. The Effect of CNMs and NIR Laser Irradiation on the Formation and Destruction of E. coli VKM B-3858D Biofilms

We studied the formation of biofilm in a BTN broth with graphene derivatives. GO and rGO were shown to significantly reduce the biomass of 1–7-day-old *E. coli* VKM B-3858D biofilms. In the presence of CNMs, by the 7th day of growth, only 38–39% of the biofilm biomass was formed compared to the control (Figure 3). The presence of graphene and its derivatives in the cultivation medium caused a significant decrease in both the polysaccharide component of the matrix and extracellular proteins (Figure 4). The greatest decrease in the content of exopolysaccharides and matrix protein was observed during biofilm growth in a medium with GO and rGO, respectively. The NIR laser irradiation of the inoculum (two periods of 1.5 min with a 45 s break) resulted to a significant decrease in biofilm formation. The greatest suppression of *E. coli* VKM B-3858D biofilm formation was observed in the presence of rGO (Figure 5a).

The metabolic rate and intracellular ATP content of mature biofilm cells significantly decreased when exposed to graphene derivatives, NIR laser irradiation alone, and when laser irradiation was performed in a medium with graphene derivatives (Figure 5b,c), with the maximum decrease observed when biofilms were irradiated with a NIR laser in the presence of graphene derivatives. A 99.6% decrease in the intracellular ATP content was achieved when biofilms were irradiated with a NIR laser in the presence of rGO.

### 2.3. The Effect of GO and rGO on Cell Membrane Permeability

The effect of GO and rGO on the permeability of the cytoplasmic membrane of *E. coli* VKM B-3858D cells was studied using several methods: epifluorescence microscopy of 18 h and 3-day biofilm cells stained with BacLightTM Bacterial Viability Kits and the concentration of ATP, nucleotides and protein released into the medium from destroyed cells of 3-day biofilms after exposure to GO and rGO. It was shown that exposing the biofilm’s cells to GO and rGO for 1 h causes an increase in the concentration of extracellular ATP, extracellular nucleotides and protein in the medium, which is a result of cytoplasmic membrane integrity disruption (Table 1). At the same time, unmodified graphene (RG-T1) did not cause an increase in membrane permeability. The 18 h biofilms incubated in 0.9% NaCl without CNMs contained mostly live cells with impermeable membranes, whereas the older 3-day biofilms contained a fairly large number of cells with permeable membranes (Figure 6). Exposure to GO and rGO for 1 h resulted in an increase in the number of cells with permeable membranes. It was shown that rGO washes away most of the biofilm from the glass surface, leaving minor areas of attached cells.

## 3. Discussion

We assessed the stress response of the bacterial cell by studying the expression of the general stress response gene *rpoS* and the oxidative stress response gene *soxS*. An increase in the expression level of *rpoS*—the gene encoding the sigma subunit of RNA polymerase—under GO, rGO, laser irradiation and a combination of these factors indicates that the cell perceives these exposures as a stress factor, which is accompanied by the activation of the cellular stress response systems. In this case, the effect can be mediated by a violation of the cytoplasmic membrane integrity. It was previously shown that MWCNTs-COOH and SWCNTs-COOH cause a reliable increase in *rpoS* expression; moreover, MWCNTs-COOH had an antioxidant effect, while SWCNTs-COOH had a prooxidant effect [39]. However, in this study, an increase in the *soxS* expression level under the influence of graphene derivatives was not demonstrated. The effect of NIR laser irradiation on the cell is not explained only by a local increase in temperature, since exposure to 45 °C for 2 min did not cause an increase in *rpoS* expression. An increase in *rpoS* expression was only noted upon heating in the presence of graphene derivatives. However, graphene derivatives did not cause an increase in the expression level of *soxS*, a gene included in the oxidative stress protection regulon. Moreover, in the presence of the strong oxidizing agent PQ, GO and rGO reduced *soxS* expression compared to that under the effect of PQ alone. This could be associated either with the adsorption of PQ by CNMs and a decrease in its effect on the cell, or with scavenging superoxide radicals by graphene derivatives when radicals are trapped in crystal lattice defects, and a decrease in the damaging effect of these ROS. The effect of a NIR laser on the cell results in an increase in *soxS* expression; however, in the presence of graphene derivatives, *soxS* expression was significantly lower than when under laser irradiation alone. There is evidence that a 1270 nm NIR laser causes the formation of singlet oxygen [40]. Data on the effect of GO on the formation of oxidative stress are ambiguous. Thus, there is evidence that GO causes ROS-independent oxidative stress [17,18,41,42]. ROS, including superoxide anion, are not formed under the effect of graphene derivatives, but graphene nanoparticles could oxidize glutathione, which serves as a mediator of the redox state in bacteria [18]. In this case, glutathione probably acts as a defense against the direct or indirect action of graphene derivatives. ROS are neutralized by glutathione, which leads to its oxidation, but *soxS* expression does not increase.

However, the oxidation of glutathione makes cells more vulnerable to subsequent exposure to oxidants. In our study, we showed that the expression level of *soxS* does not increase during the growth of *E. coli* EH40 in a medium with graphene derivatives, while the expression level of *rpoS* in *E. coli* RO91 increases, indicating the formation of a non-specific response to stress caused by graphene derivatives.

Membrane disruption under the influence of GO and rGO was shown using the method of staining cells with the LIVE/DEAD™ dye and subsequent epifluorescence microscopy, as well as determining extracellular ATP, nucleotides and protein. When comparing the effects of GO and rGO, we have shown that GO has a greater damaging effect on *E. coli* VKM B-3858D cells than rGO. The contact of cells with GO resulted in an increase in the concentration of extracellular ATP, protein, and nucleic acids by 3.5–5 times compared to the control, which indicates a violation of the membrane integrity. Oxidized groups of graphene have a stronger destructive effect on cells, interacting with membranes. A reduction in oxidized groups somewhat reduces the damaging effect of nanocarbon on the cells of gram-negative bacteria. When studying the permeability of the cytoplasmic membrane of biofilm cells, we have shown that when 18 h biofilms were treated with GO and rGO for 1 h, the number of cells with a permeable membrane significantly increased. The increased permeability of the cytoplasmic membrane may be a consequence of various factors. Oxidative stress can cause lipid peroxidation, which results in the destruction of lipopolysaccharides and the cell membrane. However, by studying the expression of *soxS,* we have shown that graphene derivatives are not the cause of ROS-dependent oxidative stress. It is known that the physical destruction of membranes is possible under the influence of graphene derivatives. The sharp edges of graphene plates can penetrate into the lipid bilayer, disrupting it. In addition, graphene can facilitate the extraction of membrane lipids [16]. In this case, the disruption of membrane permeability could have been caused by the direct destructive action of graphene derivatives. The effect of rGO on the biofilm was characterized by the more intense removal of biofilms from the glass surface. The hydrophobicity of rGO contributes to the eradication of biofilm formed on a hydrophilic surface.

CNMs also have antibacterial effects without NIR radiation [15,16,19]. In our study, we showed that GO and rGO reduce the formation of *E. coli* VKM B-3858D biofilms when grown in the presence of these CNMs dispersed in the medium. This decrease in biofilm formation may be associated with the negative effect of CNMs on cell viability and/or a decrease in the massiveness of the exopolymer matrix. We showed that the content of exogenous protein and polysaccharides in the matrix significantly decreases during biofilm growth in the presence of graphene and its derivatives. This may be due to both the negative effect on the synthesis of EPS and the extraction of already synthesized macromolecules upon contact with CNMs. The penetration of graphene nanoparticles into a cell can cause the disruption of enzyme functions, including those involved in the synthesis of exopolysaccharides, and also negatively affect membrane transport systems that facilitate the efflux of synthesized polymers from the cell. CNMs have a fairly high adsorption capacity for macromolecules. GO and other graphene derivatives are a great adsorbent material, effectively removing various metals, organic dyes and pharmaceuticals from water and binding macromolecules [43]. It is known that proteins interact with GO, and these connections can lead to alterations in the conformation and deterioration of the biological functions of proteins. In addition, the number of bound protein molecules increases for graphene with a higher degree of oxidation [44]. Thus, GO and rGO can aggregate with exopolymers, facilitate their removal from the biofilm matrix and transition to an unbound state, or disrupt the functions of enzymes involved in the synthesis and transport of exopolysaccharides. As a result, the structure of the biofilm is disrupted, it becomes less stable, and it is more easily subjected to dispersion and eradication.

It has been previously shown that carbon nanotubes, depending on their functionalization, can have either a minor anti-biofilm or a more significant pro-biofilm effect [45], while unmodified MWCNTs have a pronounced pro-biofilm effect on the formation of bacterial biofilms after preliminary adhesion and their development in the absence of a planktonic culture [46]. Graphene derivatives have a greater damaging effect on bacterial cells than pristine and functionalized MWCNTs. This is due to the structure of nanocarbon, since the sharp edges of graphene nanoplatelets damage the membrane to a greater extent than nanotubes. In addition, there is evidence that rGO interacts with the lipid layer of the membrane and causes its destruction [16], and that graphene destabilizes protein–protein interactions by disrupting hydrophobic bonds and interacts with hydrogen groups of bacterial RNA/DNA, which results in the interruption of the replication phase [9].

The effect of CNMs on the formation and destruction of mature *E. coli* VKM B-3858D biofilm was enhanced by laser irradiation (1270 nm). In this case, the most significant damaging effect on biofilms was achieved by the combination of rGO and laser irradiation: rGO absorbed NIR light to a greater extent than GO and converted it into heat. Previously, Lima-Sousa et al. reported that when irradiated with NIR light, thermogel–rGO produced a 3.8-fold higher temperature increase than thermogel–GO, thereby reducing the viability of breast cancer cells by 60% [47]. Laser irradiation of the inoculum without CNTs reduced biofilm formation by 20%. The combined effect of laser irradiation and rGO made it possible to suppress biofilm formation by 92%. Irradiation without CNMs and in the presence of rGO reduced the concentration of intracellular ATP in mature biofilms by 37% and 99.6%, respectively. Previously, Khan et al. reported the higher antimicrobial activity of GO when irradiated with near-infrared radiation (NIR laser, λ = 1064 nm) compared to GO alone [26]. Henriques et al. reported that few-layer graphene and few-layer GO exposure to low-intensity NIR killed 85% of adherent methicillin-resistant *S. aureus* and *S. epidermidis* [48]. Low concentrations of GO (2 μg/mL) and 15 min irradiating with NIR radiation killed 98.49% of *P. aeruginosa* planktonic cells [49]. We have shown that brief NIR laser irradiation (no more than 3 min) with 200 μg/mL rGO results in the death of 99.6% of *E. coli* cells in mature biofilms. As a result of our experiments, it was proven that the most complete eradication of mature biofilm and the maximum suppression of both the biofilm formation process and the viability of cells of mature biofilms can be achieved with the complex effect of 3 min irradiation with a 1270 nm laser at 5 A in the presence of 200 mg/L rGO.

The limitations of this work are related to the cytotoxicity of graphene derivatives. Although graphene oxide is used, for example, to coat implants and promote good osseointegration [50], the question of its effect on cell lines remains unanswered. Apparently, the cytocompatibility/cytotoxicity of graphene nanomaterials depends on the concentration and physical and chemical properties (hydrophobicity, number of graphene sheets, and size) [51]. It is known that the cytotoxicity of GO is higher compared to rGO [50]. Since the data on the cytotoxicity of graphene nanomaterials are quite contradictory, the implementation of methods associated with the use of graphene derivatives and NIR laser irradiation require further thorough testing for biosafety and can for now only be proposed for abiotic surfaces.

## 4. Materials and Methods

### 4.1. Strains and Cultivation Conditions

The expression of the *soxS* oxidative stress genes was studied using the genetically modified *E. coli* EH40 (GC4468 λEH40 (soxS’::lacZ)) [52], kindly provided by Bruce Demple (Stony Brook University, New York City, NY, USA). The expression of the *rpoS* gene was studied using the *E. coli* RO91 (MC4100(λRZ5:rpoS742::lacZ[hybr])), kindly provided by Regina Hengge (University of Berlin, Berlin, Germany); the strains were cultured on LB medium (VWR, Radnor, PA, USA).

Biofilm formation was studied using the *E. coli* VKM B-3858D; isolated from pig feces and possessing an increased ability to form biofilms, the strain was kindly provided by Dr. Kuznetsova (Perm Federal Research Center, Perm, Russia) [53]. The strain was cultured in BTN broth (Biotekhnovatsiya, Obolensk, Russia) containing the following (g/L): peptone-5, casein hydrolysate-5, soy flour hydrolysate-12, autolyzed yeast extract-2, NaCl-5, and potassium phosphate-1.

### 4.2. Carbon Nanomaterials and Setup for Laser Irradiation

We used the following CNMs: GO with a bulk density of 2.0–2.5 g/cm^3^, particle size of 10–100 μm, more than 80% of monolayer, and containing 46% C, 49% O, 2.5% H, and 2.5% S; rGO with a bulk density of 0.01 g/cm^3^, total specific surface area of 700 m^2^/g, particle size of 10–100 μm, more than 80% of monolayer and the following elemental composition: 96% C, 2.7% O, 0.3% H, 0.4% S, and 0.5% N; graphene nanoplatelets RG-T1 and RG-S1 with a C content of at least 99%, a thickness of 5–15 nm and 3–10 nm, a diameter of 0.1–10 μm and 0.5–10 μm, a density of less than 40 kg/m^3^ and 210 ± 50 kg/m^3^, and a specific surface area of 20–25 m^2^/g and 9–10 m^2^/g, respectively (RUSGRAPHENE, Moscow, Russia). GO and rGO were added to the culture medium to a final concentration of 200 mg/L. A two-cascade fibrous NIR laser with an output wavelength of 1270 nm was used (Appendix A).

### 4.3. Genes Expression

*E. coli* EH40 (GC4468 λEH40 (soxS’::lacZ)) and *E. coli* RO91 (MC4100(λRZ5:rpoS742:: lacZ[hybr])) were cultured for 8–10 h at 37 °C in LB broth (VWR, Radnor, PA, USA), then 50 μL of the inoculum was added to 50 mL of LB broth and cultured for 15 h in Erlenmeyer flasks at 100 rpm in a GFL 1092 thermostatic shaker (GFL, Berlin, Germany). The culture was diluted with LB broth to OD600 = 0.1 in 40 mL Erlenmeyer flasks. After reaching OD600 = 0.3, the culture was transferred to Erlenmeyer flasks containing 10 mL of LB broth and 200 mg/L GO or rGO. The mixture was pre-treated with ultrasound in an Elma Ultrasonic 30S bath, (Elma, Singen, Germany) at 37 kHz 10 times for 1 min. Methylviologen hydrate (paraquat, PQ) (Aldrich, Schnelldorf, Germany) was added at a concentration of 1 μg/mL. In all cases, the nutrient medium contained 25 μg/mL streptomycin. To study the antioxidant effect of graphene derivatives together with carbon nanomaterials, PQ was added. To study the effect of laser irradiation, 200 μL of the sample was irradiated for 1 min at 5 A before determining the activity of β-galactosidase. The expression of the *soxS* and *rpoS* genes was assessed by the activity of β-galactosidase via the Miller method based on the ability of this enzyme to hydrolyze o-nitrophenyl-β-D-galctopyranoside (ONPG) (Sigma, Cibolo, TX, USA). After collection, 200 μL of the culture was placed in test tubes in 1800 μL of Z-buffer (Na_2_HPO_4_—0.06 M; NaH_2_PO_4_—0.04 M; KCl—0.01 M; MgSO_4_—0.001 M; β-mercaptoethanol—0.05 M; pH 7.0) and treated with a mixture of 0.1% sodium dodecyl sulfate (20 μL) and chloroform (40 μL) with vigorous shaking for 10 s. The reaction was started by adding 400 μL of ONPG (4 mg/mL). The reaction was carried out for 15 min at 28 °C and stopped by adding 1 mL of 1 M Na_2_CO_3_ solution to the reaction mixture. Before measuring, 3 mL of distilled water was added to the test tubes. The color intensity was assessed using a UV 1280 spectrophotometer (Shimadzu, Kyoto, Japan) according to the optical density (OD420). CFU was assessed after seeding successive tenfold dilutions of the culture on a dense LB nutrient medium.

### 4.4. Determination of Biofilm Biomass

*E. coli* VKM B-3858D biofilms were grown for 1–7 days in wells of a 96-well plate in 200 μL of the BTN broth inoculated with 5 μL of the bacterial suspension containing (1.5–1.7 × 10^9^) CFU/mL. The BTN broth contained 200 mg/L CNMs, and the BTN broth without CNMs was a control. To obtain a homogeneous suspension, the medium with CNMs was pre-treated with ultrasound in an Elma Ultrasonic 30S bath (Elma, Germany) at a frequency of 37 kHz for 10 min. Planktonic cells were removed from the wells by decantation, the biofilm was washed twice with 200 μL of potassium phosphate buffer, and the biofilm biomass was determined. The biofilm was stained with 0.1% crystal violet for 40 min in the dark. Then, the dye was removed, the stained biofilm was washed once with potassium phosphate buffer, and the dye was extracted with 96% ethanol (200 μL). Biofilm formation was assessed by the optical density of the staining solution at 540 nm using an Infinite M1000 Pro plate reader (Tecan, Mannedorf, Switzerland). To determine the effect of laser irradiation and CNMs on biofilm formation, the *E. coli* VKM B-3858D inoculum was diluted to OD540 = 0.02 and 200 μL of the suspension was irradiated for 3 min at 5 A in a well of a 96-well plate (2 periods of 1.5 min with a 45 s break). GO or rGO was added to the suspension; the suspension without laser exposure and without CNMs was a control. In these and subsequent experiments, the biofilm was grown for 3 days. Based on the study of the dynamics of biofilm formation, it was found that the 3-day biofilm had the largest biomass and more clearly reflected the effect of graphene derivatives. *E. coli* VKM B-3858D was cultured for 3 days and the biofilm biomass was determined as described previously. The effect of laser irradiation and CNMs on biofilm destruction and eradication was assessed as follows. *E. coli* VKM B-3858D biofilms were grown in BTN broth for 3 days in wells of a 96-well plate; 200 mg/L CNMs (GO or rGO) in 200 μL of 0.9% NaCl was added to the washed biofilms and the cells were irradiated for 3 min at 5 A (2 periods of 1.5 min with a 45 s break). After 1 h, the supernatant was removed and the metabolic rate was assessed using the MTT reagent and by measuring the intracellular ATP content.

### 4.5. Determination of Metabolic Intensity

The metabolic intensity of biofilm cells was assessed by the degree of reduction of 3-(4,5-dimethyl-2-thiazolyl)-2,5-diphenyl-2-H-tetrazolium bromide (MTT, Servicebio, Wuhan, China) to water-insoluble formazan, followed by the extraction of formazan with DMSO. Then, 160 μL of 0.9% NaCl and 40 μL of MTT dye were added to the biofilms and incubated for 4 h at 37 °C. The reduction of the tetrazolium salt by cellular dehydrogenases resulted in the formation of a precipitate of purple formazan crystals, which were dissolved in DMSO, and the optical density of the solution was determined at λ 570 nm with an Infinite M1000 pro plate reader (Tecan, Mannedorf, Switzerland).

### 4.6. Determination of ATP Content

*E. coli* VKM B-3858D biofilms were washed twice with potassium phosphate buffer and 200 μL of DMSO was added. After 15 min of exposure in DMSO, the samples were frozen at −18 °C and kept the required time until determination. A lyophilized reagent containing firefly luciferin and luciferase (ATP reagent, BHM ST, Moscow, Russia) was diluted with deionized water according to the manufacturer’s protocol, then additionally diluted 25 times with deionized water and mixed in a 1:1 ratio with samples diluted 10 times with deionized water; then, 100 μL of the mixture was added to the wells of a white opaque flat-bottomed plate (Nunc, Rosklide, Denmark). The iuminescence intensity was measured with an Infinite M1000 Pro plate reader (Tecan, Mannedorf, Switzerland). The amount of ATP was determined using a calibration curve, and for biofilms, it was calculated per well of the plate.

### 4.7. Determination of Polysaccharide and Protein Components of the Biofilm Exopolymer Matrix

Inoculum of *E. coli* VKM B-3858D (100 μL) containing (1.5–1.7 × 10^9^) CFU/mL was added to 2 mL of BTN broth with 200 mg/L GO and rGO in the wells of a 24-well polystyrene plate (NEST^®^, Wuxi, China). After 3 days of cultivation at 30 °C, the medium was removed, the biofilms were washed with 0.9% NaCl once, and the concentration of exopolysaccharides and extracellular protein was determined.

To determine exopolysaccharides, 2 mL of BTN broth containing 40 μg/mL Congo red were added to the biofilms. The biofilms were treated for 10 min in an Elma Ultrasonic 30S ultrasonic bath (Elma, Germany), incubated for 20 min at 37 °C, the liquid was centrifuged for 20 min at 14,000× *g*, the supernatant was placed in the wells of a 96-well transparent polystyrene flat-bottomed plate (Medpolymer, Saint Petersburg, Russia) and optical density was measured with an Infinite M1000 Pro plate reader (Tecan, Mannedorf, Switzerland) at 490 nm. The control was the BTN broth with Congo red. The binding of Congo red to polysaccharides was determined by the difference between the control and the experiment.

To determine the extracellular protein, the biofilms were vortexed for 30 s, then incubated on ice for 10 min, repeated twice. Then biofilms were treated in an ultrasonic bath Elma Ultrasonic 30S (Elma, Germany) for 10 min, the liquid was passed through a filter with a pore diameter of 0.22 μm and mixed with ice-cold acetone in a ratio of 1 part liquid to 7 parts acetone. Centrifuged for 40 min at 14,000× *g*, the sediment was dissolved in 0.9% NaCl and mixed in equal proportions with Bradford reagent. OD was measured at 595 nm with an Infinite M1000 Pro plate reader (Tecan, Mannedorf, Switzerland), the protein concentration was calculated using a calibration graph.

### 4.8. Epifluorescence Microscopy

Biofilms of *E. coli* VKM B-3858D were grown for 18 h or 3 days in 25 mL of BTN broth inoculated with 0.5 mL of bacterial suspension containing (1.5–1.7 × 10^9^) CFU/mL on microscope slides (25 × 75 mm). To assess the effect of graphene derivatives on membrane permeability, an 18-h growth period was chosen in addition to the 3-day period, since 3-day biofilm cells lost viability when incubated in 0.9% NaCl, which made it difficult to assess the effect of graphene derivatives. Slides with biofilms were washed with 0.9% NaCl and incubated for 1 h in 0.9% NaCl with 200 mg/L GO and rGO. Then the biofilms were stained in the dark for 20 min with dyes (Syto 9/propidium iodide) LIVE/DEAD™ BacLight™ Bacterial Viability Kits (Invitrogen, Waltham, MA, USA), at the rate of 3 μL of the dye mixture in equal proportions per 1 mL of 0.9% NaCl, incubated in the dark for 20 min and visualized in a light microscope (Leica DM LS, Wetzlar, Germany) with epifluorescence. Cells with an intact and damaged membrane were stained green and red, respectively.

### 4.9. Determination of Nucleotides, ATP and Protein in the Extracellular Medium

Inoculum of the *E. coli* VKM B-3858D (100 μL) containing (1.5–1.7 × 10^9^) CFU/mL was added to 2 mL of BTN broth in the wells of a 24-well polystyrene plate (NEST^®^, China). After 3 days of cultivation at 30 °C, the medium was removed, the biofilms were washed with 0.9% NaCl once and 2 mL of 0.9% NaCl with 200 mg/L GO and rGO were added. After 1 h of incubation at 22–25 °C, the liquid was centrifuged for 20 min at 14,000× *g*, the supernatant was passed through a filter with a pore diameter of 0.22 μm and the ATP concentration was determined as described above. The concentrations of DNA, RNA and protein in the extracellular medium were determined using an analyzer (NanoDrop 2000, Thermo Fisher Scientific, Wilmington, DE, USA).

### 4.10. Statistical Analysis

The presented data are the results of three independent experiments. The results obtained were processed statistically, and the means, standard deviations, and confidence intervals were determined. The significance of differences was assessed using Student’s *t*-test, *p* < 0.05.

## 5. Conclusions

Graphene derivatives have an antibiofilm effect on *E. coli*. The effect on 24-h biofilms is less pronounced, whereas by the 7th day the biofilm biomass decreases under GO and rGO almost half. Irradiation of the *E. coli* inoculum with a 1270 nm NIR laser and irradiation in the presence of rGO reduced biofilm formation by 20 and 92%, respectively. Exposure to CNMs alone, laser irradiation alone, and the combined effect of CNMs and laser irradiation causes a decrease in both the metabolic activity of mature biofilms and the content of intracellular ATP. The maximum decrease in the intensity of metabolism and intracellular ATP content was observed in cells exposed to combined treatment with rGO and irradiation. In irradiated cells, the ATP content decreased by 37%, and when laser irradiation was combined with GO and rGO, the ATP content decreased by 90 and 99.6%, respectively. The maximum release of nucleotides and ATP from *E. coli* cells into the external environment, as well as an increase in the number of cells with a permeable membrane, is observed under the effect of GO. Exposure to rGO results in the physical removal of (washing off) biofilms from a glass. It was found that in the culture medium, graphene derivatives initiate a general stress response in bacterial cells, but do not activate the expression of the *soxRS* regulon genes. GO significantly reduces the *soxS* expression under oxidative stress induced by PQ. Irradiation (1270 nm, 1 min, 5 A) increases the level of *soxS* expression relative to non-irradiated bacteria, and the addition of rGO reduces the expression of this gene in irradiated cells. At the same time, *rpoS* expression increased when exposed to GO and rGO, laser irradiation, as well as under the combined effect of graphene derivatives and irradiation or heating (2 min 45 °C), but does not change under heating without graphene derivatives. Thus, graphene derivatives, especially rGO, in combination with NIR laser irradiation can be an effective means of eradication of gram-negative bacterial biofilms both at the stage of their formation and with mature biofilms.

Further developments may include studies of the cellular transcriptome under the influence of graphene derivatives and NIR laser irradiation, which will allow for a more detailed determination of changes in bacterial gene expression. In addition, the implementation of this approach in the clinic requires a more detailed study of the combined effect of graphene and NIR laser on cell lines, which will allow for the selection of the most cytocompatible carbon nanomaterial.

## Figures and Tables

**Figure 1 ijms-26-04728-f001:**
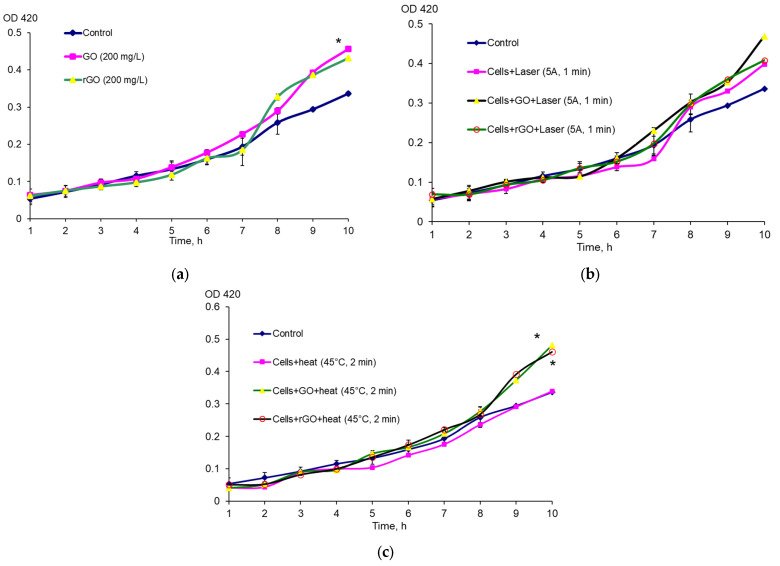
Expression of *rpoS* under the effect of CNMs (**a**), the combined effect of CNMs and NIR laser irradiation (**b**), heating and CNMs (**c**), * *p* < 0.05 (*n* = 3). *E. coli* RO91 cells without exposure served as a control. Data are presented as mean ± error of the mean.

**Figure 2 ijms-26-04728-f002:**
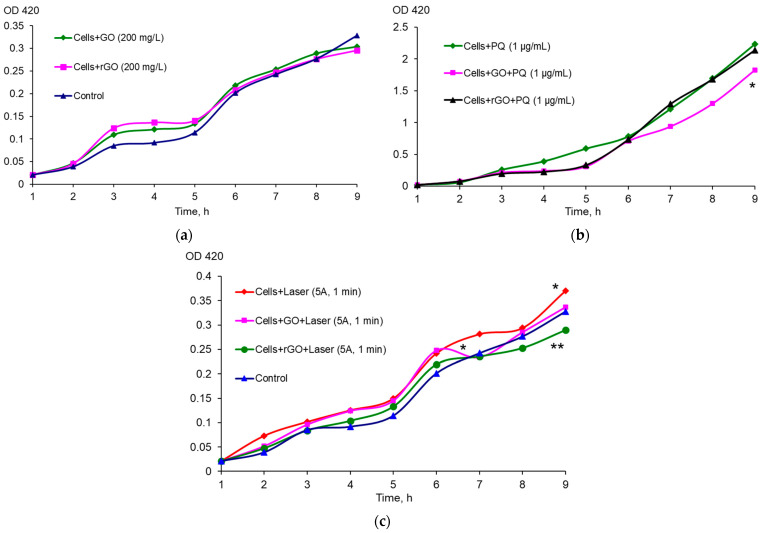
Expression of *soxS* exposed to CNMs (**a**), CNMs and PQ (**b**), combined exposure to CNMs and NIR laser irradiation (**c**), * *p* < 0.05 (*n* = 3) difference from control, ** *p* < 0.05 (*n* = 3) difference between combined exposure to NIR laser + CNMs and exposure to NIR laser alone. The control was *E. coli* EH40 cells without exposure. The data are presented as an example of one of the experiments. The trend persisted in three independent experiments.

**Figure 3 ijms-26-04728-f003:**
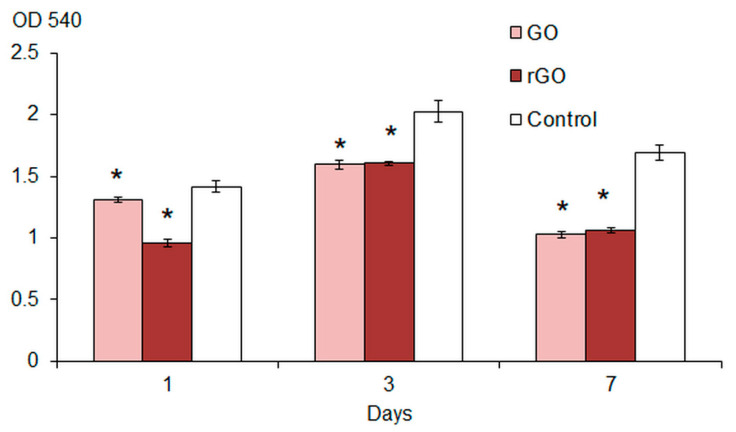
Biomass of *E. coli* VKM B-3858D biofilms during 1–7 days of growth in nutrient broth with GO, rGO and without CNMs (Control), * *p* < 0.05 (*n* = 16).

**Figure 4 ijms-26-04728-f004:**
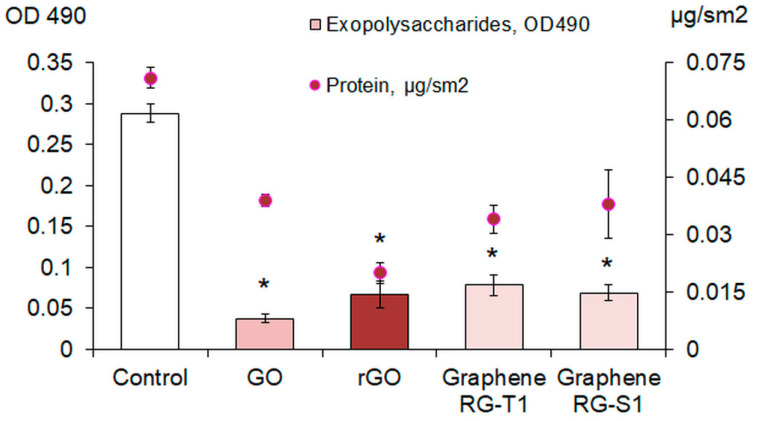
Exopolysaccharide and protein content in the matrix of *E. coli* VKM B-3858D biofilms grown for 3 days in nutrient broth without CNMs (Control) and nutrient broth with GO, rGO, Graphene RG-T1, and Graphene RG-S1, * *p* < 0.05 (*n* = 6).

**Figure 5 ijms-26-04728-f005:**
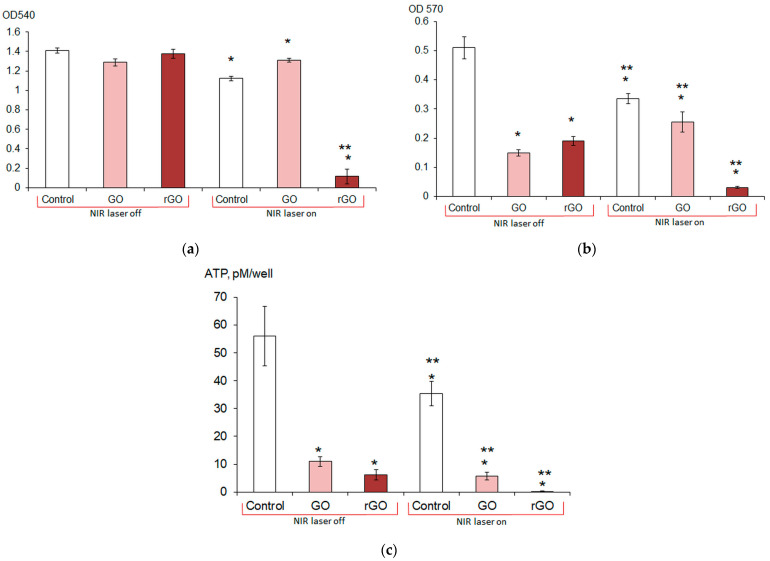
Formation of *E. coli* VKM B-3858D biofilms (**a**) and destruction of mature biofilms (**b**,**c**) without irradiation (NIR laser off) and under irradiation (NIR laser on): Control (without CNMs), exposure with GO and with rGO. * Difference from control, *p* < 0.05 (*n* = 5–16); ** difference between irradiated and non-irradiated samples, *p* < 0.05 (*n* = 5–16). Biofilm biomass was estimated by OD 540 of extracted crystal violet dye (**a**), reduction of tetrazolium salt to formazan in cells (metabolic intensity) (**b**), and intracellular ATP content (**c**).

**Figure 6 ijms-26-04728-f006:**
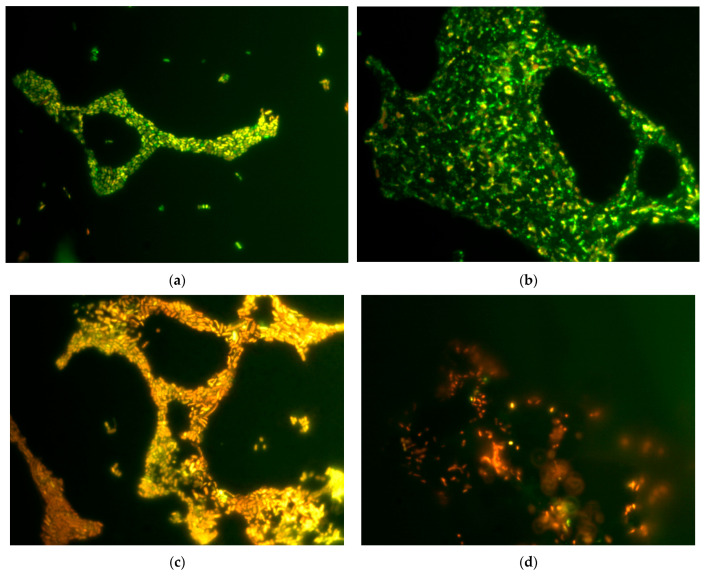
Epifluorescence microscopy of 18 h (**a**–**d**) and 3-day (**e**–**h**) *E. coli* VKM B-3858D biofilms in a nutrient broth (**a**,**e**) and after 1 h of incubation in 0.9% NaCl (**b**,**f**), 0.9% NaCl with 200 mg/L GO (**c**,**g**) and 0.9% NaCl with 200 mg/L rGO (**d**,**h**). Green color indicates living cells, red color indicates dead cells. Magnification is 1000×.

**Table 1 ijms-26-04728-t001:** Concentration of nucleotides, protein and ATP in the medium after the destruction of *E. coli* VKM B-3858D biofilm cells by graphene derivatives for 1 h.

CNMs	DNA, ng/μL	RNA, ng/μL	Protein, mg/mL	ATP, pM/mL
Control	5.85 ± 2.05	5.25 ± 1.35	0.09 ± 0.03	0.54 ± 0.39
GO	27.07 ± 15.37	21.50 ± 15.52	0.48 ± 0.29	1.70 ± 0.62
rGO	17.85 ± 5.75	15.15 ± 6.05	0.32 ± 0.12	0.70 ± 0.20
Graphene RG-T1	1.95 ± 0.65	2.35 ± 0.65	0.02 ± 0.01	0.22 ± 0.17

## Data Availability

Data is available upon request.

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
