# Peer review of "Effects of Graphene Derivatives and Near-Infrared Laser Irradiation on E. coli Biofilms and Stress Response Gene Expression"

_ijms, 2025, doi:10.3390/ijms26104728_

Round 1
Reviewer 1 Report
Comments and Suggestions for Authors
In their work, Authors investigated the combined effect of graphene oxide (GO), reduced graphene oxide (rGO), and near-infrared (NIR) laser irradiation on the formation and eradication of E. coli biofilms. The research focuses on assessing biofilm biomass reduction, cellular stress responses, and membrane integrity, highlighting the potential of GO/rGO with NIR light as an effective non-invasive strategy against Gram-negative bacterial biofilms. Although the study lacks originality, as several works have addressed similar objectives using GO or rGO for antibacterial and antibiofilm purposes in combination with other strategies—including NIR irradiation—it provides insights that could contribute to advancing knowledge in the field. The work is currently highly challenging, particularly in the following rationale, the reading of obtained results, and their interpretation/discussion.
Here are the main issues:
Introduction. It provides a clear overview of the properties of graphene and its potential antibacterial/antibiofilm applications, as well as its complex relationship with oxidative damage mechanisms. It also introduces the role of NIR irradiation, but completely lacks a descriptive overview of the metabolic pathways associated with stress response genes. Additionally, a potential cytotoxicity of GO and rGO is introduced (line 67), but it is not discussed further, nor is there any mention of testing with cell lines to indicate the proposed system's human safety.
The aim also seems to differ between the abstract and the introduction: in the abstract, the objective of the work appears to be applied, as it states, "The investigation is devoted to the effect of graphene oxide (GO), reduced graphene oxide (rGO) and NIR laser irradiation on the formation and eradication of E. coli biofilms," while in the introduction, the focus seems to be fundamental, as it states, "The aim of the study was to determine the expression level of bacterial cell stress response genes (soxS, oxidative stress response, and rpoS, general stress response) on genetically modified E. coli strains harboring gene fusions under GO, rGO and laser irradiation...".
Results. They are presented poorly and do not facilitate the reader's understanding and interpretation. Almost all the figures need clearer descriptions to guide the reader in interpreting the results. In particular, the authors should focus on improving the visual clarity of the figures. The figures are often difficult to understand and frequently require the reader to refer back to the caption or other parts of the document (for example, in Figure 2, to understand the meaning of RG, the reader must go to section 2.3). In many cases, legends, graph titles, and conditions on the X-axis are missing, with the X-axis simply labeled as 1, 2, 3... The style should also be standardized, as each figure has a different graphical style, making it difficult to compare results under the same experimental conditions.
Discussion. The paper does not provide a real discussion of the results obtained. The entire paragraph from lines 207 to 229 is not a discussion but rather a literature review on phototherapy and its use against antibiotic resistance. This could be integrated into the introduction, keeping in mind that this topic is already extensively covered. A true discussion of the data begins only at line 250, "When comparing the effects of GO and rGO…", but it does not offer any insights to the reader on, for example:
- The increase in cellular permeability in the absence of oxidative stress. In general, oxidative stress induces lipid peroxidation, which leads to the destruction of LPS and the cell membrane.
- Similarly, it does not provide any insights into the ambiguous expression of soxS, which seems to be normal in the presence of GO and rGO alone, but decreases in the presence of paraquat. The discussion is limited to just two lines: "ROS, including superoxide anion, are not formed under the effect of graphene derivatives, but graphene nanoparticles could oxidize glutathione, which serves as a mediator of the redox state in bacteria."
- The same applies to the composition of the extracellular matrix of biofilms, which is poorer in the presence of graphene and its derivatives. The results are simply restated, but there is no discussion, which is instead focused on the final antifouling and anti-biofilm effects.
The Materials and Methods. The section should also be improved, as the experimental steps chosen in relation to the results obtained earlier are not always clear. For example, why is the biofilm evaluated at 1, 3, and 7 days in paragraph 4.4, and then only at 3 days later on? Similarly, in paragraph 4.8, the biofilm is evaluated after 3 days, but the photos in Figure 6 show 18 and 3 days. In this regard, the effect of NaCl alone, observed in panel F, is interesting; it seems that there was a physiological switch inducing the morphological transition of Pseudomonas aeruginosa from a rod shape to a spherical one (see article https://doi.org/10.1371/journal.pcbi.1006012).
The conclusions are limited to a repetition of the results obtained but do not offer insights into the limitations of the study. They do not clearly propose future studies to further validate the findings or explore unknown pathways, nor do they address the challenges of applying this approach in a clinical setting.
Author Response
Reviewer 1
We thank the Reviewer for the detailed analysis of our manuscript and comments that allowed to improve the presentation of the work. The changes made to the manuscript are marked in green.
- Introduction.
We added a description of genes associated with the cell response to stress, a description of the mechanisms involved in cell adaptation to oxidative and general stress, and added references (Lines 58-70). The discussion of the possible cytotoxicity of graphene derivatives was transferred to the Discussion section (Lines 342-351). In the Introduction we removed the mention of cytotoxicity. The review of phototherapy was moved from the Discussion section to the Introduction (Lines 104-135).
- The aim
The aim in the Abstract and in the Introduction was brought to a unified form, focusing on the fundamental (Lines 16-19).
- Results.
The results were presented in the order that corresponds to the aim: first, the results associated with gene expression, then biofilm formation. All diagrams were redone, presented in a more visual form, and the appearance was standardized.
- Discussion.
The review of the literature on phototherapy and its use has been moved to the Introduction. We added discussion regarding the increase in membrane permeability in the absence of oxidative stress (Lines 285-292). The ambiguous expression of soxS is discussed by us in Lines 255-258, 261-268. We added discussion regarding EPS (Lines 304-309).
- The Materials and Methods.
This section has been updated with an explanation of why 3-day biofilms were studied in some experiments, while 18-hour and 3-day biofilms were stained by LiveDead dye (Lines 417-420, 476-479). We thank the Reviewer for drawing attention to the fact of the morphological transition of cells into a spherical shape; this may be the subject of further study.
- The conclusions
The conclusions have been updated with limitations and prospects for further research (Lines 526-531). The limitations of this work were also added at the end of the Discussion section (Lines 342-351).
Reviewer 2 Report
Comments and Suggestions for Authors
The investigation is devoted to the effect of graphene oxide (GO), reduced graphene oxide (rGO) and NIR laser irradiation on the formation and eradication of E. coli biofilms. These carbon nano-materials (CNMs) are shown to reduce biofilm biomass and the content of extracellular polymeric substances (EPS), both exopolysaccharides and protein in the biofilm matrix. GO, rGO, and laser irradiation have a negative effect on E. coli biofilms both at the cell adhesion stage and on mature biofilms, with the greatest suppressive effect being exerted by the combination of NIR laser with rGO. Laser irradiation with a wavelength of 1270 nm in the presence of GO and rGO causes suppression of the metabolic rate and a decrease in the intracellular ATP content by 94 and 99.6%, respectively. GO, rGO, NIR laser irradiation separately and irradiation in the presence of graphene derivatives causes an increase in the expression level of rpoS associated with the general stress response of bacteria. The experiments are systematic and the discussion is logical and convincing. Therefore, I propose to publish it in this journal after "minor revision".
Below are my concerns:
- In some ways, the abstract is too long. It is recommended to be concise. This paper mainly describes the key issues around this research, what work has been done and what basic conclusions have been obtained.
- There are some papers have devoted to the effect of graphene oxide and NIR laser irradiation on the formation and eradication of bacteria biofilms. For example, Graphene films irradiated with safe low-power NIR-emitting diodes kill multidrug resistant bacteria, Photothermally enhanced bactericidal activity by the combined effect of NIR laser and unmodified graphene oxide against Pseudomonas aeruginosa, A dual-targeted platform based on graphene for synergistic chemo-photothermal therapy against multidrug-resistant Gram-negative bacteria and their biofilms. The differences should be added in the introduction.
- Although the introduction of this paper has been as much as possible to outline the near infrared light induced catalytic reactions and the production of reactive oxygen species. But there are some recent studies that have been overlooked. For example, Near-Infrared Light Responsive TiO2 for Efficient Solar Energy Utilization; Strategies to extend near-infrared light harvest of polymer carbon nitride photocatalysts.
- The English of the manuscript should be polished carefully when you revise your manuscript.
- The logical correlation between paragraphs, sentences, and different chapters needs to be further strengthened.
Author Response
Reviewer 2
We thank the Reviewer for the positive assessment of our work and the critical comments that allowed to improve the manuscript. We have marked the changes made in the manuscript in blue.
- We shortened the abstract, removed less significant statements.
- We added references and information suggested by the Reviewer to the Discussion and compared data from studies "Graphene films irradiated with safe low-power NIR-emitting diodes kill multidrug resistant bacteria", "Photothermally enhanced bactericidal activity by the combined effect of NIR laser and unmodified graphene oxide against Pseudomonas aeruginosa", "A dual-targeted platform based on graphene for synergistic chemo-photothermal therapy against multidrug-resistant Gram-negative bacteria and their biofilms" with our results (Lines 333-338, 136-140).
- We added the references «Near-Infrared Light Responsive TiO2 for Efficient Solar Energy Utilization»; «Strategies to extend near-infrared light harvest of polymer carbon nitride photocatalysts» when describing photosensitizers that enhance the effect of NIR radiation (Lines 124-125).
- The English language was edited.
- To improve the logical connection, the sequence of results was changed. The manuscript presented the results in the following sequence: 1) gene expression; 2) biofilm formation; 3) EPS; 4) metabolism; 5) membrane permeability. Logical connections were added in some places of the manuscript (Lines 245-248, 26, 166).